# Encapsulation of Clofazimine by Cyclodextran: Preparation, Characterization, and In Vitro Release Properties

**DOI:** 10.3390/ijms24108808

**Published:** 2023-05-15

**Authors:** Seong-Jin Hong, Young-Min Kim

**Affiliations:** Department of Integrative Food, Bioscience and Biotechnology, Chonnam National University, Gwangju 61186, Republic of Korea; 197282@jnu.ac.kr

**Keywords:** clofazimine, cyclodextran, cyclodextrin, inclusion complex, drug release properties

## Abstract

This study aimed to evaluate and compare the efficacy of cyclodextrans (CIs) and cyclodextrins (CDs) in improving the water solubility of a poorly water-soluble drug, clofazimine (CFZ). Among the evaluated CIs and CDs, CI-9 exhibited the highest percentage of drug inclusion and the highest solubility. Additionally, CI-9 showed the highest encapsulation efficiency, with a CFZ:CI-9 molar ratio of 0.2:1. SEM analysis indicated successful formation of inclusion complexes CFZ/CI and CFZ/CD, accounting for the rapid dissolution rate of the inclusion complex. Moreover, CFZ in CFZ/CI-9 demonstrated the highest drug release ratio, reaching up to 97%. CFZ/CI complexes were found to be an effective means of protecting the activity of CFZ against various environmental stresses, particularly UV irradiation, compared to free CFZ and CFZ/CD complexes. Overall, the findings provide valuable insights into designing novel drug delivery systems based on the inclusion complexes of CIs and CDs. However, further studies are needed to investigate the effects of these factors on the release properties and pharmacokinetics of encapsulated drugs in vivo, in order to ensure the safety and efficacy of these inclusion complexes. In conclusion, CI-9 is a promising candidate for drug delivery systems, and CFZ/CI complexes could be a potential formulation strategy for the development of stable and effective drug products.

## 1. Introduction

Clofazimine (CFZ) is a riminophenazine antibiotic that was first introduced in the 1960s for the treatment of leprosy (Figure 1a) [1]. Since then, its potential has been explored in other mycobacterial infections, such as tuberculosis and *Mycobacterium avium* complex (MAC) infections, as well as in other conditions, such as certain types of cancer and autoimmune diseases [2,3]. However, its use is associated with several adverse effects, including skin and mucous membrane discoloration, gastrointestinal symptoms, ocular toxicity, and adverse cardiac effects. CFZ works by binding to the bacterial DNA and inhibiting its replication, thereby preventing the growth and spread of *Mycobacterium* [4]. It also has anti-inflammatory and immunomodulatory effects. CFZ is mainly used in the treatment of leprosy, multidrug-resistant tuberculosis (MDR-TB), and MAC infections [3]. In leprosy, it is used in combination with other antibiotics, such as dapsone and rifampicin, to shorten the duration of treatment and prevent relapse [5].

CFZ has also shown promise in the treatment of non-infectious conditions, such as certain types of cancer and autoimmune diseases, including Behcet’s disease and systemic lupus erythematosus [6,7]. CFZ is a lipophilic drug with a long half-life, and is mainly excreted in feces [8]. However, it is poorly absorbed in the gastrointestinal tract and has variable absorption rates, depending on the formulation and accompanying food intake [2]. CFZ is categorized as a Biopharmaceutics Classification System (BCS) class II drug that exhibits poor solubility (0.210 μg/mL) in aqueous media, posing significant challenges for its formulation and delivery [2]. Several strategies have been explored to improve its solubility, including the use of modified CDs, cosolvents, surfactants, and complexation agents [2,9,10] in order to enhance its bioavailability and therapeutic efficacy. Co-solvents, such as ethanol, propylene glycol, or polyethylene glycol, can increase the solubility of lipophilic drugs by facilitating their partition into the aqueous phase [9]. However, the use of co-solvents is associated with several drawbacks, including toxicity, immunogenicity, and instability. Another approach is the use of surfactants, such as Tween 80 and sodium lauryl sulfate (SLS). Surfactants can improve the solubility of lipophilic drugs by reducing their interfacial tension and increasing their dispersibility. However, surfactants can also cause adverse effects, such as irritation, inflammation, and surfactant-induced lung injury. Complexation agents, including cyclodextrins and solid dispersions, have also been explored for CFZ solubilization [11].

Cyclodextrin (CD), a cyclic oligosaccharide with *α*-1,4-glycosidic bonds, possesses a unique ability to form inclusion complexes with a wide range of hydrophobic molecules, including drugs, dyes, and flavors (Figure 1b–d) [12]. This ability has made CDs attractive in drug delivery systems, as they can improve the solubility, stability, and bioavailability of drugs [13,14]. One of the most significant benefits of CDs is their entrapment ability, which involves the formation of host–guest complexes that can protect the entrapped molecule from degradation, increase its solubility, and improve its bioavailability. The entrapment ability of CDs is based on their hydrophobic cavity, which allows the hydrophobic guest molecule to fit inside and form a stable complex. The size of the CD cavity and the shape of the guest molecule are critical factors for the entrapment efficiency of CDs. The ability of CDs to form inclusion complexes with hydrophobic molecules is influenced by several factors, such as the concentration of the CD, the pH in the medium, and the temperature. CDs have been widely used in various drug delivery systems, including nanoparticles, microparticles, and hydrogels, due to their excellent entrapment ability. The use of CDs in drug delivery systems can enhance the stability and bioavailability of drugs while reducing their toxicity and side effects [15,16]. Furthermore, the entrapment ability of CDs can be exploited to control drug release by altering the size, concentration, and structure of the CDs. However, previous studies have reported that natural CDs such as *α*-CD, *β*-CD, and *γ*-CD have different properties, where *α*-CD, compared with *β*-CD and *γ*-CD, has a lower encapsulation capacity; *β*-CD exhibits lower water solubility, and *γ*-CD is susceptible to digestion [16,17].

Cyclodextran (CI) is a circular oligosaccharide produced from dextran by the action of cycloisomaltooligosaccharide glucanotransferase (CITase) (Figure 1e–g) [18]. In this research, we discuss the unique features of Cis in comparison to CDs, and their potential applications in various industries [19]. Initially, CIs were thought to be useful in generating inclusion complexes similar to CDs, but it was later discovered that CIs possess cariostatic ability in addition to their inclusion complex ability [20]. Unlike CDs, CIs have a flexible structure containing *α*-1,6-glucosidic linkages, resulting in a 100-fold higher solubility in aqueous solutions. They also have a wider molecular diameter and exhibit better inclusion capability than CDs. CIs are known to increase bioavailability by encapsulating insoluble substances, and to improve the functionality of functional and bioactive ingredients as solubilizing agents [19]. Additionally, CIs inhibit biofilm-forming enzymes, reducing glucan and dextran synthesis, and preventing dental plaque production and periodontal disease [20]. CIs show potential as novel encapsulation agents for hydrophobic pharmaceutical biomaterials, and as alternatives to CD with a lower aqueous solubility.

Therefore, to compare the solubilization of CFZ, the physical properties and in vitro release characteristics of natural CI and CD complexes were evaluated to facilitate the application of CFZ complexes with increased solubility.

## 2. Results and Discussion

### 2.1. Phase Solubility Test

The efficacy of CFZ inclusion complexes based on CD derivatives has previously been described against two different TB strains (*M. avium*/*M. intracellulare*) [2,10,21]. Thus, the use of CIs and CDs to perform a CFZ phase solubility test was considered the first-choice option. The inclusion complexes between CFZ and CIs (CI-7, CI-8, or CI-9) and those between CFZ and CDs (*α*-CD, *β*-CD, or *γ*-CD) were successfully obtained to increase the water solubility of CFZ. CI-9 exhibited the highest percentage of drug inclusion (Figure 2a). This result was unexpected, because in previous studies, other water-insoluble drugs have shown higher affinities for more hydrophobic CDs such as β-CD.

The stability constants (*K*_α_) for complexes formed by CIs and CDs were calculated next to determine which complex provided the highest percentage of recovered CFZ. The CFZ/CI-9 complex exhibited a *K*_α_ of 0.2136, with a CFZ:CI-9 molar ratio of 1:5. For comparative purposes, the CE and the molar ratio were also determined for the CFZ/CI and CFZ/CD complexes. Compared to the CI-9, *α*-CD showed, as expected, a lower *K*_α_, and the CFZ: *α*-CD molar ratio was 2-fold lower than the corresponding molar ratios obtained with the CFZ/CI complexes. Figure 1a shows the phase solubility curves represented as dissolved drug concentrations against the concentration of CIs and CDs used. CFZ/CI-9 complex had the highest solubility, and no significant differences were detected among CFZ/CI-8, CFZ/CI-7, CFZ/*β*-CD, and CFZ/*γ*-CD. The CFZ/*α*-CD complex exhibited the lowest solubility, and the CFZ:α-CD molar ratio was determined to be 1:10.

### 2.2. Encapsulation Efficiency

The encapsulation efficiencies (EEs) of the inclusion complexes of CFZ with CIs and CDs ranged from 50.8 ± 4.52% to 91.9 ± 1.44%, revealing significant differences (Figure 2b). The highest EE (91.9%) was obtained with the CFZ/CI-9 complex, with a CFZ:CI-9 molar ratio of 0.2:1, while the lowest EE (50.8%) was observed in the case of the CFZ/α-CD complex, with a CFZ:α-CD molar ratio of 0.01:1. The differences in EE could be attributed to various factors, including the chemical structure and physical properties of the guest molecule, the type of interaction between CFZ and the CIs or CDs, and the procedure used to synthesize the complexes. According to their EEs, the CFZ/CI and CFZ/CD complexes had the following relationship: CFZ/CI-9 > CFZ/CI-8 > CFZ/β-CD > CFZ/CI-7 > CFZ/γ-CD > CFZ/α-CD. The results suggest that there are favorable interactions between CI-9 and molecules with chlorophenyl group and –NH functions. On the other hand, CI-7, α-CD, and γ-CD showed unfavorable interactions with CFZ. These findings are consistent with those reported by de Castro et al., (2020), who investigated EE and clathrate complexes at a molar ratio of 1:7 based on the complexes of CFZ and β-CD derivatives by applying the methods of spray-drying and freeze-drying [10].

Overall, these results indicate that the EEs of inclusion complexes of CFZ with CIs and CDs are influenced by various factors, including the chemical structure of the guest molecule, the type of interaction between the guest and the encapsulation agent, and the method of complex synthesis. The inclusion complex CI-9 showed the highest EE, while *α*-CD showed the lowest. These findings provide insights into the design and development of novel drug delivery systems based on the inclusion complexes of CIs and CDs. Further studies are needed to investigate the effects of these factors on the release properties and pharmacokinetics of encapsulated drugs in vivo.

### 2.3. Physical Properties of CFZ/CI and CFZ/CD Complexes

#### 2.3.1. FT-IR Analysis

Figure 3a presents the FT-IR spectra of CFZ, the CFZ/CI complexes, and the CFZ/CD complexes. The FT-IR spectrum of CFZ displayed characteristic peaks, including an N-O stretching frequency at 1500–1550 cm^−1^, N-H bending at 1625 cm^−1^, and weak N-H stretching at 2800–3000 cm^−1^, consistent with previous reports [10]. The FT-IR spectra of pure CIs and CDs showed characteristic peaks at 3316, 2919, 1347, 1132, and 1010 cm^−1^. The peaks observed at 3316 and 2919 cm^−1^ were attributed to the O-H stretching vibration in the glucose unit of CIs and CDs. Moreover, strong peaks were detected at 1347, 1132, and 1010 cm^−1^, which were assigned to the H-O-H, C-O, and C-H stretching modes, respectively. The spectra of the CFZ/CI and CFZ/CD complexes exhibited broader peaks at 1639 cm^−1^ compared to that of CFZ, suggesting the presence of N-H bending and C-N stretching in CFZ. Due to the amorphous nature of entrapped CFZ, the peak intensity could be reduced or overlapped by major constituents of CIs and CDs. The similarity between the spectra of the CFZ/CI and CFZ/CD complexes indicated the compatibility of all the formulation’s components, consistent with previous studies reporting similarities between the FT-IR spectra of CDs [10]. Hence, FT-IR analysis should be considered in conjunction with other spectroscopic and thermal analyses for an appropriate characterization of the inclusion complexes.

#### 2.3.2. X-ray Diffraction Analysis

The complexation of CFZ results in changes in its diffractogram, which can be observed as shifts in signal intensity or peak positions compared to isolated molecules or inclusion complexes, as shown in Figure 3b. The diffractogram of CFZ raw material corresponds to the polymorphic form of the drug (9.2°, 13.4°, 19.8°, and 21.7°), as reported in the Cambridge Structural Database identifier code DAKXUI03 [22]. The XRD patterns of the three CIs revealed characteristic peaks between 2θ = 18.1° to 19.1°, while the three CDs exhibited main peaks at 2θ = 10.4°, 17.8°, 20.6°, and 28.4°. This pattern is associated with the molecular arrangement of the CD, which can form small domains with short-range order. In the CFZ/CI and CFZ/CD inclusion complexes, the absence of an amorphous halo at 9.2°, 13.4°, 19.8°, and 21.7° indicates complex formation through the interaction of CFZ with CIs and CDs. The main amorphous halo of the CFZ/CI complexes ranged from 17.37° to 23.57°, while the CFZ/CD complexes exhibited new peaks at 6.5°, 11.5°, 13.6°, 17.5°, and 19.7°. These results suggest successful encapsulation of CFZ in Cis and CDs, which is consistent with previous studies [10]. To accurately characterize the inclusion complexes, XRD results should be interpreted in combination with thermal and spectroscopic analyses.

#### 2.3.3. DSC Analysis

DSC can be used for the recognition of inclusion complexes. As a guest molecule is embedded into CD cavities, its melting, boiling and sublimation points are either shifted or simply disappear [23]. The DSC of CDs showed a single endothermic peak centered at 115 °C, which is associated with the release of water from the *β*-CD [24]. Figure 4a shows the DSC of Cis as a single endothermic peak centered at 90 °C, which is associated with the release of water from the Cis. The DSC of CFZ, on the other hand, exhibited a broad endothermic peak with a maximum of 226 °C, which accords with its boiling point (212 °C). Regarding this behavior, it is important to note that the endothermic process begins around 40 °C, which tallies with the high volatility of the active compound. On the other hand, Figure 3a summarizes the different thermograms corresponding to the inclusion complexes of the CFZ/CI and CFZ/CD. The CFZ/CI and CFZ/CD complexes had the same melting peak as the endothermic peak of CFZ, remaining weakly visible at 120 °C, suggesting that CFZ was encapsulated in CIs and CDs. In the inclusion complexes, it was possible to observe a small endothermic peak at 290 °C, evidencing the presence of the CFZ. Therefore, in the same thermogram, it is possible to observe two endothermic peaks, which can be associated with the evaporation of internal and external water molecules bound to CIs and CDs or the elimination of included water molecules with different strengths of interactions with the cyclodextrin [25,26]. Moreover, the thermograms of the inclusion complexes show a complete disappearance of the peak of CFZ, which would indicate molecular encapsulation of the active agent within the cavity of the CIs and CDs, and not merely a physical mixture. This becomes evident when the thermograms of the physical mixture and the inclusion complexes are compared [27,28]. Overall, these results indicate that the active components were protected inside the cavities of the CIs and CDs.

#### 2.3.4. TGA

According to Zhu et al. (2014), TGA is a useful method to investigate alterations in physical and chemical characteristics of materials [29]. In this study, TGA was employed to examine the behavior of CFZ, CFZ/CI complexes, and CFZ/CD complexes (Figure 4b). The CFZ/CI and CFZ/CD complexes exhibited mass loss in three zones. The initial zone below 100 °C exhibited an 8% decrease in mass, which can be attributed to the removal of surface water associated with CIs and CDs. The subsequent process occurred around 200 °C and exhibited a 3% mass loss caused by the evaporation of internal water [30]. Finally, a third process at around 250–550 °C, which is related to the degradation of the CIs and CDs, was observed [31,32]. At this point, the mass was reduced by 81.4%. Furthermore, CFZ demonstrated a distinctive process at 300 °C, leading to a rapid reduction in mass. Conversely, in the TGA of the various encapsulation agents, it was evident that the thermogram associated with the CFZ/CI complex differs significantly from that of the CFZ/CD complex.

#### 2.3.5. Stability Analysis

We evaluated the stability of free CFZ, CFZ/CI complexes, and CFZ/CD complexes under the conditions of UV-induced degradation and FeCl_3_-induced oxidative degradation to assess the ability of CIs and CDs to protect the drug from environmental stresses. To this end, we measured the amounts of active CFZ in the CFZ/CI and CFZ/CD complexes over a 6 h exposure period (Figure 5a). As a control, we also included free CFZ, CFZ/CI complexes, and CFZ/CD complexes in the analysis. A significant degradation (>40%) of free CFZ was observed during the first hour of UV exposure, followed by consistent degradation of up to ~80% over a 6 h exposure. The stability of CFZ in CFZ/CI and CFZ/CD complexes under photodegradation conditions was slightly higher than that of free CFZ. However, CFZ/CD complexes also exhibited a severe degradation (approximately 70%) after 6 h of exposure. Conversely, the effect of UV irradiation on CFZ in CFZ/CI-9 was largely reduced, with less than 50% of CFZ degraded over the course of the 6 h exposure period.

In order to evaluate the ability of CIs and β-CDs to protect CFZ from oxidative degradation, we used ferric chloride (FeCl_3_) as an oxidizing agent (Figure 5b). Based on the oxidative degradation analysis, it was found that the rate of decomposition of CFZ was higher in the CFZ/CD complexes compared to the CFZ/CI complexes, indicating that complexation of CFZ with *α*-CD, and *γ*-CD is not enough to decrease the stability of CFZ against oxidative stress. Our results suggested that CFZ/CIs, and CFZ/*β*-CD provide an effective means of protecting the activity of CFZ against various environmental stresses that could take place during manufacturing, processing, storage, and consumption. Therefore, CFZ/CIs could be a potential formulation strategy for the development of stable and effective drug products. However, further studies are required to optimize the formulation and assess the safety and efficacy of the inclusion complexes for therapeutic applications.

#### 2.3.6. SEM Analysis

The morphological disparities between the individual constituents and the inclusion complex were highlighted through SEM images (Figure 6). Pure CFZ (Figure 6a) exhibited a standard state and a rectangular shape with angles. Conversely, significant changes in the morphology and shape of particles were observed in the CFZ/CI and CFZ/CD inclusion complexes (Figure 6b–g). Specifically, the original morphology of the two pure compounds was lost in the complex, while an amorphous powder with a slightly reduced size and no differentiation of the individual components were observed in all complex samples. These findings could potentially account for the observed rapid dissolution rate of the inclusion complex. The alterations in the innate morphology of the individual components suggested successful encapsulation between CFZ and CFZ/CIs and CDs, indicating the successful formation of an inclusion complex.

### 2.4. Drug Release Properties

#### 2.4.1. Drug Release Analysis

The drug release behaviors of CFZ/CI and CFZ/CD composites were analyzed using a dialysis bag (Figure 7a). Six different composites, CFZ/CI-7, CFZ/CI-8, CFZ/CI-9, CFZ/*α*-CD, CFZ/*β*-CD, and CFZ/*γ*-CD, were investigated to assess their drug release properties. CFZ/CI-9 exhibited the highest drug release rates, reaching up to 97%, while CFZ/CI-7 and CFZ/CI-8 showed drug release rates of up to 90%. These findings suggested that CI-9 was the most effective in releasing and transporting CFZ, and that *β*-CD had a high encapsulation capacity but a reduced drug release rate due to the low solubilization capacity of the encapsulant. On the other hand, CFZ/*α*-CD demonstrated the lowest drug release rate, which may be related to entrapment efficiency and solubilization properties. These findings were consistent with those reported in previous studies on CFZ composites using *β*-CD and its derivatives, which exhibited excellent trapping capacity but reduced drug release characteristics [3]. Additionally, CI-7 and CI-8 showed a continuous increase in drug release as the treatment duration increased, with release characteristics similar to those of *β*-CD, indicating that the solubilizing ability of the aggregate and the bonding force with CFZ had a significant impact on the release characteristics. Furthermore, the drug release characteristics were found to be related to the phase solubility test and entrapment efficiency.

#### 2.4.2. In Vitro Simulated Drug Release Analysis

Encapsulation is a widely used technique to improve the bioavailability and therapeutic efficacy of drugs. In the present study, the release of encapsulated CFZ from CIs and CDs was investigated in a simulated gastrointestinal environment to evaluate the effect of CIs and CDs on drug release kinetics. First, the release of CFZ from CIs and CDs in simulated saliva fluid (SSF, pH 7.0) was determined (Figure 7b), and it was found that the release of CFZ was negligible in SSF due to the protective effect of the encapsulation agents. The protective effect of CIs and CDs against the highly acidic environment of the stomach was also evaluated in simulated gastric fluid (SGF, pH 2.0). The release of CFZ from the CIs in SGF was also negligible, indicating that CIs would provide an effective barrier to protect the encapsulated CFZ against the highly acidic stomach environment. On the other hand, the release characteristics of the CFZ/CD complexes in the simulated stomach increased with time, which was attributed to the decomposition of some CDs and the weakening of the bond between CFZ and CDs. This result suggested that the CDs might not be effective as a protective barrier against the highly acidic environment of the stomach. Subsequently, the CFZ/CI and CFZ/CD complexes were incubated with simulated intestinal fluid (SIF, pH 7.0) to observe the sustainable release of CFZ.

Interestingly, all CIs with CFZ showed sigmoidal release profiles, along with a phase exhibiting a rapid release of CFZ at the early stage of the intestinal environment. The sigmoidal release profile of CFZ/CIs indicated a controlled drug release behavior. In contrast, the CFZ/CD complex exhibited a sigmoidal release profile with the slow release of CFZ in the initial stages in the intestinal environment. The transition of the release behavior from concave–downward to a sigmoidal shape was attributed to the fact that CFZ/CI complexes have a higher degree of crystallinity and a denser matrix compared to those of CFZ/CD complexes. This difference in the structural characteristics prompted drug release in a more diffusion-controlled manner.

Berkland et al., (2002) reported that the degree of crystallinity and the matrix density of the encapsulation agents affect drug release kinetics [33]. The higher degree of crystallinity and the denser matrix of CFZ/CI complexes compared to those of CFZ/CD complexes are consistent with the results of the present study. The structural disruption of CFZ/*γ*-CD over the course of the digestion reaction was detected using SEM analysis. In addition, the exceptionally low release rates of CFZ/*α*-CD in the intestinal environment suggested that the low emission characteristics affected the crystallization of CFZ due to the weakening of the bonding structure. On the other hand, CFZ/CI complexes had a stable sigmoidal release profile, suggesting that along with stable delivery, the decomposition of the aggregate did not proceed during the digestion process, and the complex remained water-soluble. The results of this study indicate that the release properties of CFZ/CI and CFZ/CD complexes would provide an effective means of delivering the guest molecules of interest in a more predictable and reproducible manner.

## 3. Materials and Methods

### 3.1. Materials and Chemicals

Cyclodextrins (*α*-CD, *β*-CD, and *γ*-CD; CDs), deuterium oxide (D_2_O), and clofazimine (CFZ) were purchased from Sigma-Aldrich (St. Louis, MO, USA). Cyclodextrans (CI-7, CI-8, and CI-9; CIs) were synthesized using CITase in our laboratory [34].

### 3.2. Preparation of the CFZ/CI and CFZ/CD Complexes

The CFZ/CIs and CFZ/CDs complexes were prepared as described previously with slight modifications [10]. The CFZ/CIs and CFZ/CDs complexes were prepared by adding an excess amount of CFZ to mixtures of ethanol and water (1:1, *v*/*v*) containing CIs (CI-7, CI-8, or CI-9) and CDs (*α*-CD, *β*-CD, or *γ*-CD). The samples were subjected to ultrasonic treatment for 30 min twice and stirred continuously at 40 °C for 48 h until equilibrium was reached. The resulting solution was filtered through a 0.45-µm pore-size filter membrane and freeze-dried. All samples used in the experiments were CFZ/CIs and CFZ/CDs that were produced after lyophilization.

### 3.3. Phase Solubility Test

The method was used to conduct a phase solubility study [35]. Initially, a range of aqueous solutions of 0–10 mM CIs (CI-7, CI-8, or CI-9) and CDs (*α*-CD, *β*-CD, or *γ*-CD) were prepared, followed by the addition of an excess amount of CFZ in 10-mL amber tubes with caps. These suspensions were then treated twice ultrasonically for 30 min, and stirred continuously at room temperature for 48 h until an equilibrium was reached. After filtration through a 0.45-μm membrane filter, the filtrates were diluted and analyzed using a UV spectrophotometer (HIDEX-sense 425-301, Hidex, Turku, Finland) at 495 nm. All experiments were repeated three times. The phase solubility diagram was used to calculate the apparent stability constants (*Kα*) of the CFZ/CI and CFZ/CD inclusion complexes using Equation (1).
*K*_α_ = slope/S_0_ (1 − slope)(1)

Here, S_0_ represents the intrinsic aqueous solubility of CFZ at 25 °C in the absence of CIs and CDs, while the slope is obtained from the straight line in the phase solubility diagram.

### 3.4. Encapsulation Efficiency (EE)

The CFZ content inside the CIs and CDs was determined by suspending 100 mg of either the CFZ/CI or CFZ/CD complex in 20 mL phosphate buffer for 24 h with constant stirring. Absorbance was measured using a UV/visible spectrophotometer at 495 nm to determine the CFZ content. All experiments were carried out at room temperature. The encapsulation efficiencies of the CFZ/CI and CFZ/CD complexes were calculated using Equation (2).
(2)$$EE=\frac{\mathrm{total}\;\mathrm{amount}\;\mathrm{of}\;\mathrm{CFZ}\;-\mathrm{free}\;\mathrm{amount}\;\mathrm{of}\;\mathrm{CFZ}}{\mathrm{total}\;\mathrm{amount}\;\mathrm{of}\;\mathrm{CFZ}}\times 100\;$$

### 3.5. Physical Properties

#### 3.5.1. FT-IR Analysis

The Fourier transform infrared (FT-IR) spectra of the CFZ/CIs and CFZ/CDs complexes were acquired using a PerkinElmer Spectrum 400 spectrophotometer (Waltham, MA, USA). The sample spectra were recorded over the range of 400–4000 cm^−1^.

#### 3.5.2. X-ray Diffraction

To examine the crystal structure of the CFZ, CFZ/CI complexes, and CFZ/CD complexes, X-ray diffraction (XRD) analysis was conducted using an X-ray diffractometer (X’Pert PRO MPD, PANalytical, Almelo, The Netherlands) equipped with Cu- *K*_α_ radiation (λ = 0.154 nm) at 50 kV and 50 mA. The XRD patterns were acquired over a 2θ range of 5–90° at a scan speed of 1° per min.

#### 3.5.3. Differential Scanning Calorimetry (DSC)

The compound’s formation was confirmed by conducting a DSC analysis. Approximately 2 mg of CFZ, CFZ/CI, and CFZ/CD were placed in an aluminum pan and heated from 25 °C to 300 °C at a scanning rate of 10 °C per min, under a nitrogen atmosphere, using a DSC instrument (DSC3, Mettler Toledo, GmbH, Greifensee, Switzerland).

#### 3.5.4. Thermogravimetric Analysis (TGA)

Thermal characterization of the CFZ, CFZ/CI complexes, and CFZ/CD complexes was performed using a thermogravimetric analyzer (TGA2, Mettler Toledo, GmbH, Greifensee, Switzerland) in a N_2_ atmosphere. The samples were heated from 20 °C to 600 °C at a rate of 10 °C per min.

#### 3.5.5. Stability Test

We induced the photodegradation of the CFZ, CFZ/CI complexes, and CFZ/CD complexes using a UV lamp (F4T5BLB, Sankyo Denki Co. Ltd., Kanagawa, Japan) and measured their stability under chemical oxidation. An aliquot of 100 μL was taken from the test sample at each time point and mixed with 400 μL of 99% DMSO to release encapsulated CFZ from the CFZ/CI and CFZ/CD complexes into the test solution [36]. The samples were positioned at a 10 cm distance from the UV lamp and exposed to a UVA wavelength range of 315–400 nm in dark at room temperature (25 ± 2 °C) for 6 h. The absorbance of the samples was measured at 492 nm using a UV-vis spectrophotometer (HIDEX-sense 425-301, Hidex, Turku, Finland) at 1 h intervals. The stabilities of the free CFZ in DMSO (0.02% *w*/*v*) and the CFZ/CI and CFZ/CD complexes in deionized water (0.02%, *w*/*v*) against photodegradation were examined as controls.

The stabilities of the CFZ, CFZ/CI complexes, and CFZ/CD complexes under chemical oxidation conditions were also tested, as described previously, with slight modifications [37]. Briefly, 1 mL of sample containing 10 mg of the CFZ, CFZ/CI complex, or CFZ/CD complex was exposed to 100 μM FeCl_3_ at room temperature with rotation for 6 h. Aliquots (100 μL) were taken from the solution at indicated time points, and the amount of active CFZ in CFZ/CI and CFZ/CD complexes was determined by measuring the absorbance at 495 nm using a UV-vis spectrophotometer (HIDEX-sense 425-301, Hidex, Turku, Finland). The stability of free CFZ in DMSO (0.02% *w*/*v*) and CFZ/CI and CFZ/CD complexes in DDW (0.1%, *w*/*v*) under chemical oxidation conditions were also tested as a control.

#### 3.5.6. Scanning Electron Microscopy (SEM)

A scanning electron microscope (JSM-IT300, JEOL, Tokyo, Japan) was used to examine the surface morphology of the samples. After washing, each sample was placed on a copper station using a conductive adhesive and subjected to vacuum-drying. Subsequently, the samples were sputter-coated with gold–palladium to enhance their electrical conductivity under low vacuum at a voltage of 20 kV.

### 3.6. Drug Release Properties

#### 3.6.1. Drug Release

The release properties of CFZ were examined using the dialysis bag method, as described by Popat et al. (2014) [38]. CFZ/CI and CFZ/CD complexes equivalent to 1 mg of CFZ were suspended in 1 mL of 0.1 M phosphate buffer (pH 7.4) with 0.5% sodium lauryl sulfate (SLS) and placed in a dialysis bag with a molecular weight cutoff of 10 kDa. The dialysis bag was then immersed in 9 mL of 0.1 M phosphate buffer with 0.5% SLS at 37 °C with continuous stirring. Samples of 100 µL were withdrawn at specific time intervals and replaced with an equal volume of fresh dissolution medium to maintain the total volume at a constant level. The curcumin content in the samples was measured at 495 nm using a UV-Vis spectrophotometer (HIDEX-sense 425-301), and the concentration of CFZ was determined using a standard curve.

#### 3.6.2. In Vitro Simulated Digestion Release Analysis

Simulated saliva fluid (SSF), simulated gastric fluid (SGF), and simulated intestinal fluid (SIF) were prepared as reported previously with slight modifications [39]. To initiate digestion, 50 mg of CFZ/CI or CFZ/CD complex was placed in a 125 mL flask and stirred continuously in a static model at 37 °C. The samples were subjected to sequential digestion: 10 mL of simulated saliva solution was added, mixed for 5 min, and a 1 mL aliquot was collected. Next, 10 mL of SGJ was added and stirred constantly, and 1 mL aliquots were collected after 30 min and 1 h of incubation. Finally, 10 mL of SIJ was added, mixed constantly, and 1 mL aliquots were collected after 2 and 4 h. All aliquots were centrifuged at 6238× *g* for 5 min, and the supernatant liquid was filtered through a 0.45 μm membrane filter. The aliquot samples were stored frozen at −80 °C until further analysis. The release of CFZ was determined using a spectrophotometer by measuring the absorbance at 495 nm.

### 3.7. Statistical Analysis

The experiments were conducted in triplicate (n = 3), and the data obtained were subjected to statistical analysis using SPSS version 23.0 (SPSS Inc., Chicago, IL, USA). A one-way analysis of variance was performed followed by Duncan’s multiple range test. Differences with *p* values lower than 0.05 were considered significant.

## 4. Conclusions

This study aimed to compare the efficacy of CFZ inclusion complexes (CIs and CDs) in increasing the water solubility of CFZ. CFZ is a poorly water-soluble drug; thus, its formulation requires the use of an effective drug delivery system. We found that CI-9 had the highest percentage of drug inclusion and provided the highest solubility among the CIs and CDs evaluated. Additionally, CI-9 showed the highest EE, with a CFZ:CI-9 molar ratio of 0.2:1. In contrast, α-CD showed the lowest EE and a molar ratio of CFZ that was 2-fold lower compared with those of the CIs. Our findings indicated that the EE of the inclusion complexes of CFZ with CIs and CDs was influenced by several factors, including the chemical structure of the guest molecule, the type of interaction between the guest and the encapsulation agent, and the method of complex synthesis. CI-9 provided the most significant advantage over the other CIs and CDs, indicating that CI-9 could potentially be a promising compound for drug delivery systems, based on the inclusion complexes of CIs and CDs. However, further studies are required to investigate the effects of these factors on the release properties and pharmacokinetics of encapsulated drugs in vivo. We also evaluated the stability and drug release properties of CFZ in CIs and CDs. The results showed that CFZ/CI complexes provide an effective means of protecting the activity of CFZ against various environmental stresses, particularly UV irradiation, compared to free CFZ and CFZ/CD complexes. Moreover, CFZ in the CFZ/CI-9 complex exhibited the highest drug release ratios, reaching up to 97%, while the CFZ/α-CD complex showed the least favorable drug release properties. SEM analysis also suggested that CFZ/CIs and CDs were successful in forming inclusion complexes, potentially accounting for the observed rapid dissolution rate of the inclusion complex. In conclusion, CFZ/CI complexes could be a potential formulation strategy for the development of stable and effective drug products. Our findings provide insight into the design of novel drug delivery systems based on the inclusion complexes of CIs and CDs. However, further studies are needed to optimize their formulation and to assess the safety and efficacy of these inclusion complexes for therapeutic applications. The results of this study indicate that CI-9 provided the most significant advantage over other CIs and CDs in terms of drug solubility and encapsulation efficiency, making it a promising candidate for drug delivery systems. Nevertheless, the release properties and pharmacokinetics of encapsulated drugs in vivo should be investigated further to ensure the safety and efficacy of these inclusion complexes.

## Figures and Tables

**Figure 1 ijms-24-08808-f001:**
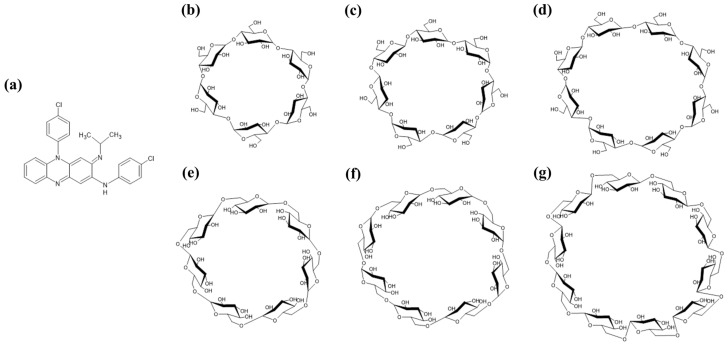
Chemical structure of CFZ, CDs, and CIs. CFZ, clofazimine; CI, cyclodextran; CD, cyclodextrin. (**a**) CFZ, (**b**) *α*-CD, (**c**) *β*-CD, (**d**) *γ*-CD, (**e**) CI-7, (**f**) CI-8, and (**g**) CI-9.

**Figure 2 ijms-24-08808-f002:**
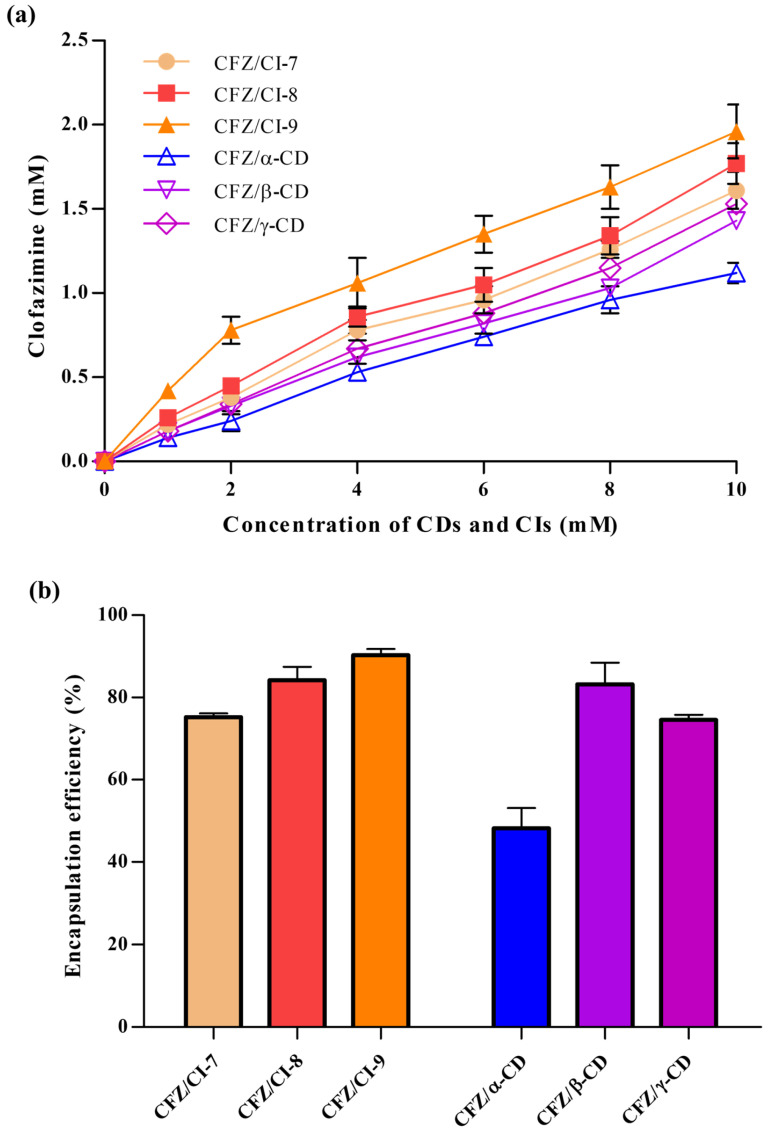
(**a**) Phase solubility test results and (**b**) encapsulation efficiencies of CFZ/CI complexes, and CFZ/CD complexes. CFZ, clofazimine; CI, cyclodextran; CD, cyclodextrin. The CIs and CDs analyzed were CI-7, CI-8, CI-9, *α*-CD, *β*-CD, and *γ*-CD.

**Figure 3 ijms-24-08808-f003:**
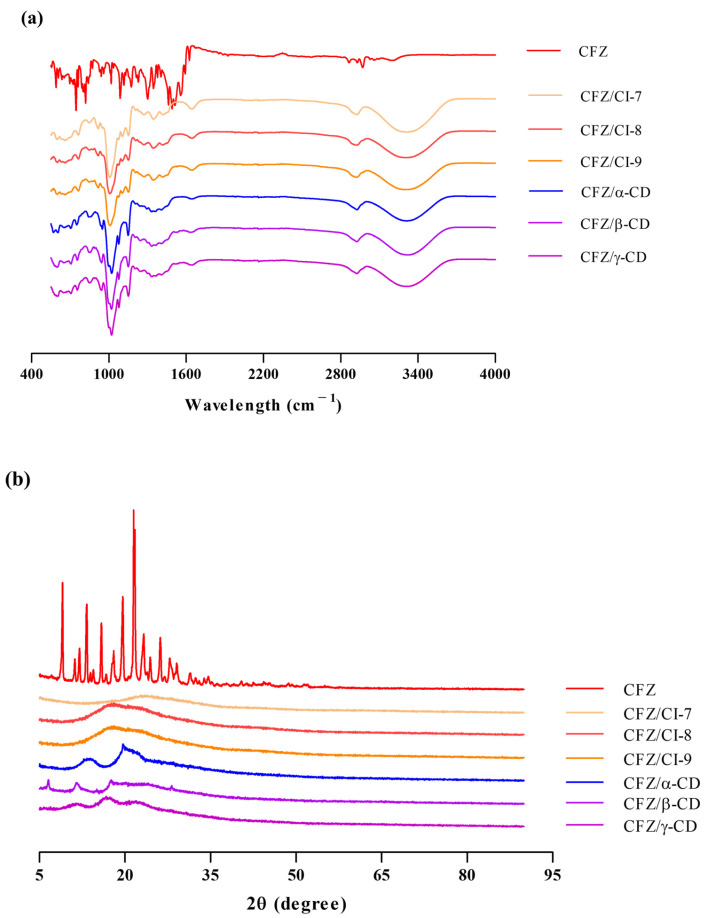
(**a**) FT-IR spectra and (**b**) X-ray diffractograms of CFZ, CFZ/CI complexes and CFZ/CD complexes. CFZ, clofazimine; CI, cyclodextran; CD, cyclodextrin. The CIs and CDs analyzed were CI-7, CI-8, CI-9, *α*-CD, *β*-CD, and *γ*-CD.

**Figure 4 ijms-24-08808-f004:**
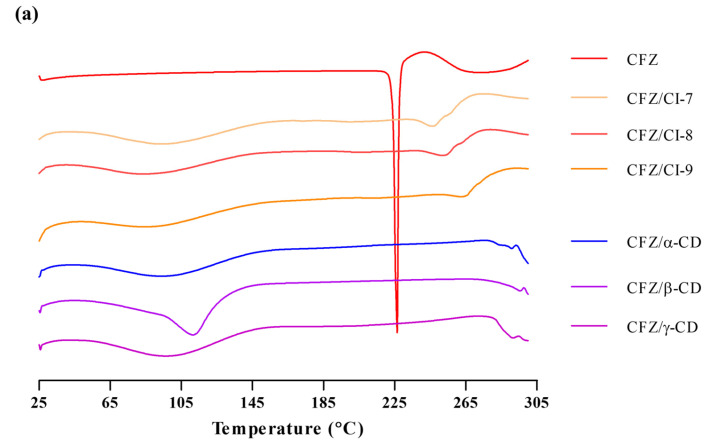
(**a**) Differential scanning calorimetry curve and (**b**) thermogravimetric analysis of CFZ, CFZ/CI complexes, and CFZ/CD complexes. CFZ, clofazimine; CI, cyclodextran; CD, cyclodextrin. The CIs and CDs analyzed were CI-7, CI-8, CI-9, *α*-CD, *β*-CD, and *γ*-CD.

**Figure 5 ijms-24-08808-f005:**
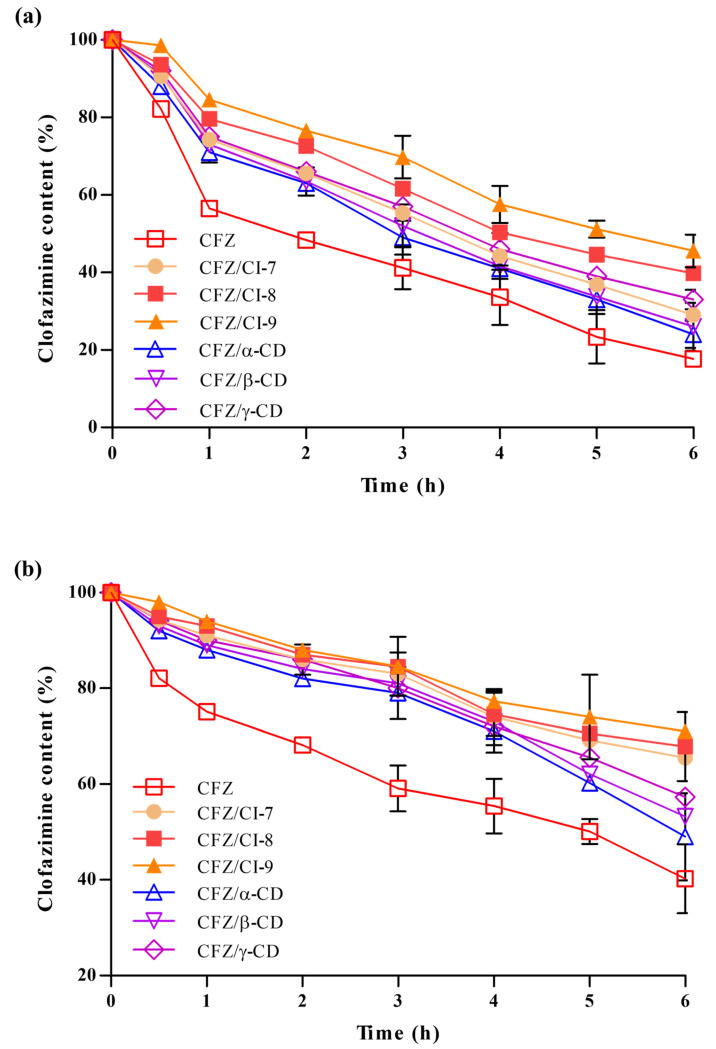
(**a**) Photodegradation and (**b**) chemical oxidation stability analyses of CFZ, CFZ/CI complexes, and CFZ/CD complexes. CFZ, clofazimine; CI, cyclodextran; CD, cyclodextrin. The CIs and CDs analyzed were CI-7, CI-8, CI-9, *α*-CD, *β*-CD, and *γ*-CD.

**Figure 6 ijms-24-08808-f006:**
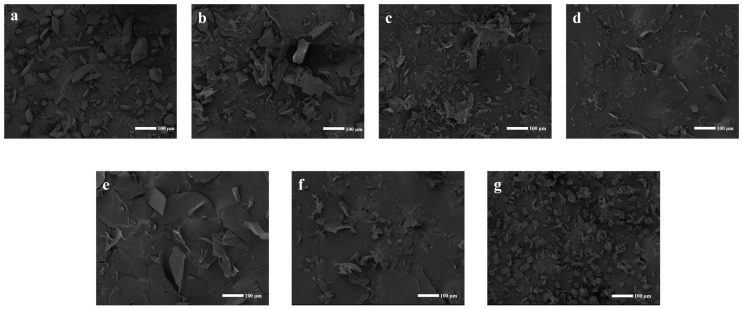
Scanning electron micrographs of CFZ, CFZ/CI complexes, and CFZ/CD complexes. (**a**) CFZ. (**b**) CFZ/CI-7. (**c**) CFZ/CI-8. (**d**) CFZ/CI-9. (**e**) CFZ/*α*-CD. (**f**) CFZ/*β*-CD. (**g**) CFZ/*γ*-CD.

**Figure 7 ijms-24-08808-f007:**
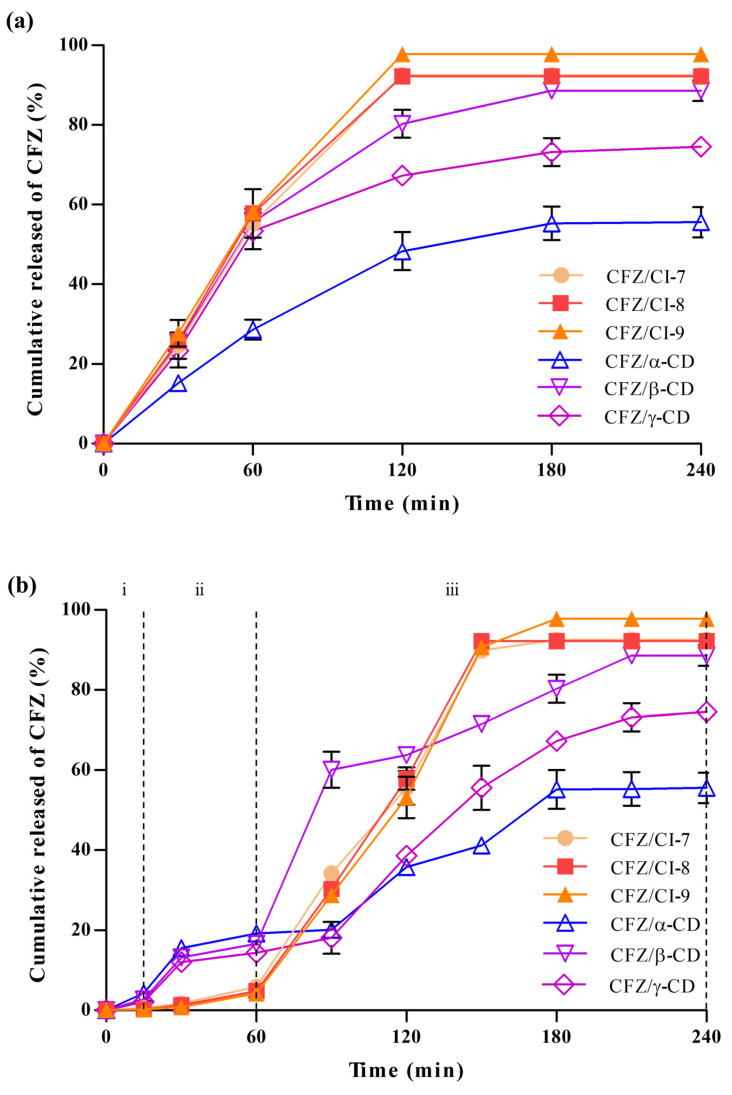
(**a**) Drug release properties and (**b**) in vitro simulated drug release properties of CFZ/CI and CFZ/CD complexes. CFZ, clofazimine; CI, cyclodextran; CD, cyclodextrin. The CIs analyzed were CI-7, CI-8, and CI-9; the CDs, *α*-CD, *β*-CD, and *γ*-CD. (i) oral phase (SSF, 5 min, pH 7.0), (ii) gastric phase (SGF, 2 h, pH 2.0), and (iii) intestinal phase (SIF, 2 h, pH 7.0).

## Data Availability

The data presented in this study are available on request from the corresponding authors.

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
