# Peer review of "Encapsulation of Clofazimine by Cyclodextran: Preparation, Characterization, and In Vitro Release Properties"

_ijms, 2023, doi:10.3390/ijms24108808_

Round 1

Reviewer 1 Report

In this submitted manuscript (ijms-2400825), Dr. Kim and Seong-Jin Hong studied the preparation, characterization, and in vitro release properties of the encapsulated clofazimine (CFZ) in cyclodextrins (CDs) and cyclodextrans (CIs). SEM, FT-IR, DSC, and TGA were applied in the characterization of sole CFZ and CDs, CIs inclusion complexes. Among these candidates tested here, the CI-9 shows the best drug inclusion and highest solubility, with an optimal encapsulation efficiency of 0.2:1 with the molar ratio between CFZ and CI-9.

It is important to investigate the effects of different vehicles for drugs in the aspect of aqueous solubility, and the findings reported in this manuscript are promising. It would appeal to the broad readership of the International Journal of Molecular Sciences, however, some of the key experiments were missed in the current report. Based on that, major revisions are recommended before further consideration.

1. In the Abstract section, it would be better to provide the full name of CFZ on line 9 as it appears for the first time.

2. It is recommended to add a scheme (in the Introduction section) to show the chemical structures of CFZ, three CDs, and three CIs that were used in this report.

3. What does CE mean on line 106? There is no explanation for this item.

4. On lines 126-127, the authors listed the EE relationships between the inclusion complexes studied in this report. Are their EE values obtained under the same conditions or are they picked from the highest values under their specific optimal conditions?

5. It is confusing to read “On the other hand, CI-7, α-CD, and γ-CD showed unfavorable interactions with β-CD” on lines 128-129. In the report, it is obvious that the authors didn’t discuss the interactions between CI-7, α-CD, γ-CD, and β-CD. It is the interaction between CFZ and the different carriers/hosts. Would the β-CD that the authors mentioned in that sentence be CFZ?

6. One reference is needed to support the conclusion on lines 151-152 for the sentence “consistent with previous studies reporting similarities between the FT-IR spectra of CIs and CDs”.

7. The formula of FeCl3 on line 215 should be revised to make the number 3 a subscript.

8. The ferric chloride was selected as the oxidizer for testing the oxidative degradation stability. Is there any specific reason to choose it compared with other oxidizers?

9. On lines 231-233, the sentence is unclear by saying “The oxidative degradation of free CFZ, CFZ/CI complexes, and CFZ/CD complexes were significantly faster than that of CFZ in CD complexes.” How could the CFZ/CD complexes be able to compare with themselves? It makes no sense to say “CFZ/CD complexes degrade significantly faster than CFZ in CD complexes”. Or in other words, are CFZ/CD complexes the same as CFZ in CD complexes? The authors should give clear definitions and descriptions for them.

10. It is not safe to draw a conclusion like this sentence on lines 235-236: “CFZ/CIs provide an effective means of protecting the activity of CFZ against various environmental stresses”. As under the oxidative condition, β-CD shows a better-protecting effect than CI.

11. In the SEM analysis section, the authors mentioned that “The morphological disparities between the individual constituents, physical mixture, and inclusion complex were highlighted through SEM images (Figure 5)”. Is there any image belonging to the samples that are “physical mixtures”? The reviewer doubts that in the whole manuscript, no characterization was done for the physical mixtures, even though the authors claimed that in several different sections, like on line 441.

12. It is hard to understand the sentence on line 251: “…suggested an interaction between CFZ and CFZ/CIs and CDs,…”. What does it mean by saying the interaction between CFZ and CFZ/CIs?

13. It is incorrect to say “CI-9 was the most effective in capturing and…” on line 263. This section discussed the releasing effect of CI-9, not the capturing effect.

14. For Fig 6b, what is the environment for testing? The authors discussed three different environments, SSF, SGF, and SIF. Which of them is the testing condition for Fig 6b? Please provide the figure legends with accurate descriptions.

15. It is hard to follow and understand the sentence “…followed by the addition of an excess amount of CFZ to separate phosphate buffer solutions” on lines 334-335. What does it mean “separate”? How does the addition of excess CFZ separate the phosphate buffer solutions?

16. Is the “DDW” on line 384 deionized water? Please provide its full name.

17. The word “CFZ” in the first sentence of the Conclusions section is not needed. Otherwise, it would make this sentence grammatically wrong.

18. The conclusion section is recommended to be rewritten. It is tedious and duplicated with the abstract and introduction sections.

19. The α-CD, β-CD, and γ-CD should have italic α, β, and γ.

For the quality of the English language, please refer to the comments provided in the above section.

Author Response

Q1. In the Abstract section, it would be better to provide the full name of CFZ on line 9 as it appears for the first time.

Author response: We changed “clofazimine (CFZ)” P1, L9.

Q2. It is recommended to add a scheme (in the Introduction section) to show the chemical structures of CFZ, three CDs, and three CIs that were used in this report.

Author response: We added the figure in CFZ, three CDs, and three CIs at P20.

Q3. What does CE mean on line 106? There is no explanation for this item.

Author response: We changed “CE” to “Kα” P3, L106.

Q4. On lines 126-127, the authors listed the EE relationships between the inclusion complexes studied in this report. Are their EE values obtained under the same conditions or are they picked from the highest values under their specific optimal conditions?

Author response: We obtained the EE values under the same conditions

Q5. It is confusing to read “On the other hand, CI-7, α-CD, and γ-CD showed unfavorable interactions with β-CD” on lines 128-129. In the report, it is obvious that the authors didn’t discuss the interactions between CI-7, α-CD, γ-CD, and β-CD. It is the interaction between CFZ and the different carriers/hosts. Would the β-CD that the authors mentioned in that sentence be CFZ?

Author response: We changed “β-CD” to “CFZ” P4, L129.

Q6. One reference is needed to support the conclusion on lines 151-152 for the sentence “consistent with previous studies reporting similarities between the FT-IR spectra of CIs and CDs”.

Author response: We changed the sentences “FT-IR spectra of CIs and CDs” to “FT-IR spectra of CDs [10]. In a previous study, the FT-IR spectra of pure CIs were determined by our research team. However, due to the fact that the results were not yet published and were undergoing revision in another journal, the FT-IR spectra of CIs were not included in the present study.

Q7. The formula of FeCl3 on line 215 should be revised to make the number 3 a subscript.

Author response: We changed “FeCl3” to “FeCl3” P8, L215.

Q8. The ferric chloride was selected as the oxidizer for testing the oxidative degradation stability. Is there any specific reason to choose it compared with other oxidizers?

Author response: Oxidants exist in various forms. However, the reason why I used ferric chloride was that it was considered one of the most exposed compounds during manufacture processing, so I conducted a stability study using ferric chloride.

Q9. On lines 231-233, the sentence is unclear by saying “The oxidative degradation of free CFZ, CFZ/CI complexes, and CFZ/CD complexes were significantly faster than that of CFZ in CD complexes.” How could the CFZ/CD complexes be able to compare with themselves? It makes no sense to say “CFZ/CD complexes degrade significantly faster than CFZ in CD complexes”. Or in other words, are CFZ/CD complexes the same as CFZ in CD complexes? The authors should give clear definitions and descriptions for them.

Author response: We changed the sentences as followings at P10, L231-233.

(After correction)

Based on the oxidative degradation analysis, it was found that the rate of decomposition of CFZ was higher in the CFZ/CD complexes compared to the CFZ/CI complexes, indicating that complexation of CFZ with α-CD, and γ-CD is not enough to decrease the stability of CFZ against oxidative stress.

Q10. It is not safe to draw a conclusion like this sentence on lines 235-236: “CFZ/CIs provide an effective means of protecting the activity of CFZ against various environmental stresses”. As under the oxidative condition, β-CD shows a better-protecting effect than CI.

Author response: We changed the sentences as followings at P10, L235-236

(After correction)

Our results suggested that CFZ/CIs and CFZ/β-CD provide an effective means of protecting the activity of CFZ against various environmental stresses that could take place during manufacturing, processing, storage, and consumption.

Q11. In the SEM analysis section, the authors mentioned that “The morphological disparities between the individual constituents, physical mixture, and inclusion complex were highlighted through SEM images (Figure 5)”. Is there any image belonging to the samples that are “physical mixtures”? The reviewer doubts that in the whole manuscript, no characterization was done for the physical mixtures, even though the authors claimed that in several different sections, like on line 441.

Author response: We confirmed that there was a problem with the representation and fixed it. Inclusions, not physical mixtures, are the correct term, and therefore we have replaced physical mixtures with inclusions complex.

We changed the “physical mixture” to “inclusion complex” as followings at P7, L182 and P10, L242.

Q12. It is hard to understand the sentence on line 251: “…suggested an interaction between CFZ and CFZ/CIs and CDs,…”. What does it mean by saying the interaction between CFZ and CFZ/CIs?

Author response: We changed “interaction” to “successful encapsulation” P10, L251.

Q13. It is incorrect to say “CI-9 was the most effective in capturing and…” on line 263. This section discussed the releasing effect of CI-9, not the capturing effect.

Author response: We changed “capturing” to “releasing” P11, L263.

Q14. For Fig 6b, what is the environment for testing? The authors discussed three different environments, SSF, SGF, and SIF. Which of them is the testing condition for Fig 6b? Please provide the figure legends with accurate descriptions.

Author response: We added the sentences as followings at P12, L277-278.

(After correction)

Figure 6. (a) Drug release properties and (b) in vitro simulated drug release properties of CFZ/CI and CFZ/CD complexes. CFZ, clofazimine; CI, cyclodextran; CD, cyclodextrin. The CIs analyzed were CI-7, CI-8, and CI-9; the CDs, α-CD, β-CD, and γ-CD. ⅰ) oral phase (SSF, 5 min, pH 7.0), ⅱ) gastric phase (SGF, 2 h, pH 2.0), and ⅲ) intestinal phase (SIF, 2 h, pH 7.0)

Q15. It is hard to follow and understand the sentence “…followed by the addition of an excess amount of CFZ to separate phosphate buffer solutions” on lines 334-335. What does it mean “separate”? How does the addition of excess CFZ separate the phosphate buffer solutions?

Author response: We revised the sentences at P13, L334-335.

(After correction)

Initially, a range of aqueous solutions of 0–10 mM CIs (CI-7, CI-8, or CI-9) and CDs (α-CD, β-CD, or γ-CD) were prepared, followed by the addition of an excess amount of CFZ in 10-mL amber tubes with caps.

Q16. Is the “DDW” on line 384 deionized water? Please provide its full name.

Author response: We changed “DDW” to “deionized water” P11, L263

Q17. The word “CFZ” in the first sentence of the Conclusions section is not needed. Otherwise, it would make this sentence grammatically wrong.

Author response: We changed the sentences as followings at P16, L432.

Q18. The conclusion section is recommended to be rewritten. It is tedious and duplicated with the abstract and introduction sections.

Author response: We changed the sentences as followings at P16, L432-.

(After correction)

This study found that inclusion complexes (CIs and CDs) could effectively increase the water solubility of CFZ. Among the evaluated CIs and CDs, CI-9 exhibited the highest percentage of drug inclusion and provided the highest solubility, as well as the highest encapsulation efficiency. CFZ/CI complexes were found to be effective in protecting the activity of CFZ against various environmental stresses and showed favorable drug release properties. Overall, CI-9 could be a promising candidate for drug delivery systems based on inclusion complexes of CIs and CDs, although further studies are needed to optimize the formulation and assess the safety and efficacy for therapeutic applications.

Q19. The α-CD, β-CD, and γ-CD should have italic α, β, and γ.

Author response: We changed “α-CD, β-CD, and γ-CD” to “α-CD, β-CD, and γ-CD”.

Reviewer 2 Report

This article presents the results of the studies on the complexes formed between an antibiotic, clofazimine, and various CDs and CIs. Although the studied complexes with CDs have been deeply studied recently, comparison with the CIs complexes is something new. The study is generally well designed and presented but I’ve also detected some major and minor methodological issues. I hope the Authors can improve and revise their valuable manuscript.

In the introduction, the figure presenting the chemical structure of CFZ should be introduced

Line 42, what is its BCS Class? Also, what is its solubility in water (g/ml)?

Line 45, have the CDs or CIs used for this purpose before? If yes, those previous studies should be described in more details here as well.

Lines 72-75, while this is true, the most important aspect is the size of their cavities. This has a major influence on their complexation affinities towards various guest. Usually, small guests are being better complexed by α while large by γ. From my experience, β is the best choice if you have no idea whether a particular guest can form a complex with a CD at all.

Line 357, how many scans were performed? Also, have you used ATR or KBr?

Line 163, the refcode must be provided.

Line 167, what do you mean by “main diffractogram”?

Line 168, I wouldn’t call them peaks as they are significantly overlapped by the amorphous halo. It’s clear that all of the complexes are amorphous.

A very important aspect: for such comparisons the FT-IR spectra and PXRD diffractograms of pure (non-complexed) CDs (α, β, γ) and CIs (7, 8, 9) must be recorded and presented.

Figure 5, the scale is not clear, this white stripe is 100 um, am I right? This figure must be improved. Also, please consider increasing the sizes of the images.

English is OK.

Author Response

Comments and Suggestions for Authors

Ref.No.: ijms-2400825

Title: Encapsulation of clofazimine by cyclodextran: Preparation, characterization, and in vitro release properties

Basically, we revised the manuscript by following important comments from reviewers.

We want to express the deeply appreciation to Reviewers for improving our manuscript. Corrected part colored as a red (suggested by reviewer #1), blue (suggested by reviewer #2).

Reviewer #2

This article presents the results of the studies on the complexes formed between an antibiotic, clofazimine, and various CDs and CIs. Although the studied complexes with CDs have been deeply studied recently, comparison with the CIs complexes is something new. The study is generally well designed and presented but I’ve also detected some major and minor methodological issues. I hope the Authors can improve and revise their valuable manuscript.

In the introduction, the figure presenting the chemical structure of CFZ should be introduced

Q1. Line 42, what is its BCS Class? Also, what is its solubility in water (g/ml)?

Author response: We changed the sentences as followings at P1, L42-45.

(After correction)

CFZ is categorized as a Biopharmaceutics Classification System (BCS) class II drug that exhibits poor solubility (0.210 μg/mL) in aqueous media, posing significant challenges for its formulation and delivery [2].

Q2. Line 45, have the CDs or CIs used for this purpose before? If yes, those previous studies should be described in more details here as well.

Author response: We changed the sentences as followings at P2, L45-46.

Q3. Lines 72-75, while this is true, the most important aspect is the size of their cavities. This has a major influence on their complexation affinities towards various guest. Usually, small guests are being better complexed by α while large by γ. From my experience, β is the best choice if you have no idea whether a particular guest can form a complex with a CD at all.

Author response: Thank you in advance for your advice. Although studies have been conducted on the complexity of guest molecules according to the size of the inner diameter, it is not unconditionally those small guest molecules are better complicated by α-CD. Therefore, three types of CD were used, and β-CD captures CFZ well, but it is difficult to use because of its low water solubility.

Q4. Line 357, how many scans were performed? Also, have you used ATR or KBr?

Author response: A total of 3 times scans were performed and the ATR mode was used.

Q5. Line 163, the ref code must be provided.

Author response: We changed the sentences as followings at P6, L163.

(After correction)

The diffractogram of CFZ raw material corresponds to the polymorphic form of the drug (9.2°, 13.4°, 19.8°, and 21.7°) as reported in the Cambridge Structural Database identifier code DAKXUI03 [22].

Q6. Line 167, what do you mean by “main diffractogram”?

Author response: We changed “main diffractogram” to “main amorphous halo” P6, L169.

Q7. Line 168, I wouldn’t call them peaks as they are significantly overlapped by the amorphous halo. It’s clear that all of the complexes are amorphous.

Author response: We changed “peaks” to “amorphous halo” P6, L168.

Q8. A very important aspect: for such comparisons the FT-IR spectra and PXRD diffractograms of pure (non-complexed) CDs (α, β, γ) and CIs (7, 8, 9) must be recorded and presented.

Author response: We added the sentences as followings at P5, L148-153, and P6, L170-172.

Q9. Figure 5, the scale is not clear, this white stripe is 100 um, am I right? This figure must be improved. Also, please consider increasing the sizes of the images.

Author response: We change the scales as following at P10, L256.

Round 2

Reviewer 1 Report

In the revised manuscript (ijms-2400825-v2), the authors have provided sufficient and detailed discussions based on the reviewers' comments. Corresponding revisions were also made to improve the quality of the manuscript. With that, I'm happy to recommend accepting it in its current version.

Reviewer 2 Report

The Authors have answered all my questions and revised their manuscript accordingly. This version can be accepted for publication.